# Functional Outcome in Spinal Meningioma Surgery and Use of Intraoperative Neurophysiological Monitoring

**DOI:** 10.3390/cancers14163989

**Published:** 2022-08-18

**Authors:** Christopher Marvin Jesse, Pablo Alvarez Abut, Jonathan Wermelinger, Andreas Raabe, Ralph T. Schär, Kathleen Seidel

**Affiliations:** Department of Neurosurgery, Inselspital, Bern University Hospital, University of Bern, 3010 Bern, Switzerland

**Keywords:** spinal meningioma, meningioma, intraoperative neurophysiological monitoring, IDEM, McCormick scale, functional outcome

## Abstract

**Simple Summary:**

Spinal meningiomas are among the most common intradural spinal tumors. Although most of them are benign, they can cause serious neurological impairment due to spinal cord compression resulting in myelopathy. The main treatment is microsurgical gross total resection. So far, only a few small case series on functional outcome and the value of intraoperative neurophysiological monitoring have been published. In our single-center retrospective cohort, we analyzed functional outcome after surgical treatment with or without intraoperative neurophysiological monitoring. We found that preoperative functional impairment is associated with tumor extension. Most patients improved after surgical resection. The recurrence rate was low and perioperative complications were in the range reported in the published studies. Intraoperative neurophysiological monitoring was helpful in challenging cases and resulted in a change of operative strategy. These findings suggest that surgical treatment of spinal meningiomas is safe and that use of intraoperative neurophysiological monitoring is advisable in complex cases.

**Abstract:**

Data on intraoperative neurophysiological monitoring (IOM) during spinal meningioma (SM) surgery are scarce. The aim of this study was to assess the role of IOM and its impact on post-operative functional outcome. Eighty-six consecutive surgically treated SM patients were included. We assessed pre and post-operative Modified McCormick Scale (mMCS), radiological and histopathological data and IOM findings. Degree of cord compression was associated with preoperative mMCS and existence of motor or sensory deficits (*p* < 0.001). IOM was used in 51 (59.3%) patients (IOM-group). Median pre and post-operative mMCS was II and I, respectively (*p* < 0.001). Fifty-seven (66.3%) patients showed an improvement of at least one grade in the mMCS one year after surgery. In the IOM group, only one patient had worsened neurological status, and this was correctly predicted by alterations in evoked potentials. Analysis of both groups found no significantly better neurological outcome in the IOM group, but IOM led to changes in surgical strategy in complex cases. Resection of SM is safe and leads to improved neurological outcome in most cases. Both complication and tumor recurrence rates were low. We recommend the use of IOM in surgically challenging cases, such as completely ossified or large ventrolateral SM.

## 1. Introduction

Spinal tumors are classified as either extradural or intradural depending on their location in relation to the spinal meninges. Intradural tumors can be further subcategorized into extramedullary or intramedullary spinal tumors [1]. Even though spinal meningiomas (SM) are rare, they are among the most frequent intradural extramedullary tumors, representing about 1.2–12% of all meningiomas. SM constitute one-third of primary spinal tumors and 25–46% of all intradural spinal tumors [1,2,3,4,5,6,7,8,9,10,11].

SM are mostly benign tumors with a peak age incidence between 40 and 70 years and a male to female ratio of 1:4–9. Most SM are WHO grade 1, with higher grades (2 to 3) reported in 1.5–8.5% of cases [7,10,11,12,13,14,15,16]. Their distribution along the spinal axis varies, but most (67–84%) are located within the thoracic region [1,2,3,4,6]. The incidences of SM in the cervical and lumbar region are 14–27% and 2–14%, respectively [1,5,16].

SM are slow-growing tumors and become symptomatic as soon as their mass effect begins to compromise the integrity of the spinal cord and nerve roots [5]. Close to 50% of SM patients complain of back pain. Other symptoms are radicular pain, motor deficits and sensory loss. In advanced disease, there can be dissociated long tract signs, including Brown-Sequard syndrome [2]. For many patients, the diagnosis is not confirmed until neurological deficits or gait disturbances become apparent [5,6].

The primary treatment option is microsurgical resection, which aims to achieve complete tumor removal and spinal cord decompression, thus preventing further neurological deterioration [1,4,17,18]. Follow-up (FU) imaging is also required. Local tumor control is usually good, with recurrence rates lower than 10% [3,7,19,20].

Intraoperative neurophysiological monitoring (IOM) is used in spine surgery for continuous real-time assessment of the neural structures at risk (spinal cord and nerve roots) and aims to reduce post-operative neurological deficits. IOM provides the opportunity for continuous evaluation of the sensory and motor functions of the spinal cord using somatosensory evoked potentials (SSEP) and motor evoked potentials (MEP) [21,22,23,24,25,26,27].

Advancements in minimally invasive techniques necessitate studies of larger cohorts of patients undergoing open microsurgical techniques for future comparison [1]. Furthermore, a study of long-term outcomes is necessary to determine the optimum treatment strategy [4].

We describe 86 cases of SM surgically treated in our institution with and without IOM. The aims of this study were to determine the functional outcome post-operatively, to analyze the clinical features, to determine prognostic factors, and to evaluate the influence of IOM on changes in surgical strategy and on functional outcome.

## 2. Material and Methods

### 2.1. Study Design and Patient Selection

We conducted a retrospective, single-center, observational cohort study. Consecutive adult patients who underwent surgery for SM in our institution between January 2009 and April 2021 were included. Patients who did not undergo surgical treatment or had solely cranial meningioma were excluded. Approval from the local ethics committee of the canton Bern, Switzerland, was obtained for this study. All patients included in this study followed the general consent procedure permitting the use of health-related data.

### 2.2. Data Analysis

Clinical data, details of intraoperative findings and procedures, and post-operative clinical outcomes were obtained from our electronic patient and clinical information system and transferred to a database. Extent of surgical resection was graded on a scale of I to V according to the Simpson grading system [28]. Whereas grade I signified complete resection including dura and bone, grade V was assigned for simple decompression with biopsy. We defined Simpson grade I to III resection as gross total resection. Histopathological data (e.g., WHO grade, ossification or calcification and subtype) were extracted from the neuropathological reports. Pre and post-operative muscle strength were assessed on a scale from 0 to 5 using the British Medical Research (BMR) Council classification [29]. For analyses, we recorded the muscle strength of the most inhibited movement. Pre and post-operative status were assessed using the Modified McCormick Scale (mMCS) (grades I–V; I = neurologically intact; II = mild motor or sensory deficit, functionally independent; III = moderate deficit, independent with external aid; IV = severe motor or sensory deficit, care required; and V = paraplegic or quadriplegic) [30] and the Frankel scale (grades A–E; A = complete; B = sensory only; C = motor useless; D = motor useful; E = no neurological deficit/complete recovery) [31] after 1 year or at latest FU if FU was less than 1 year.

### 2.3. Surgical Technique and Follow-Up

Except for one patient with a ventral meningioma at C1–2, who was operated on from a lateral approach, all patients were operated on via a posterior approach. Surgery was performed with patients in the prone position under general intravenous anesthesia with muscle relaxation for intubation purposes only. Following installation of neurophysiological monitoring equipment [21] and intraoperative localization of the level of interest, a posterior approach was used. Depending on the extent and location of the tumor, a full laminectomy or hemilaminectomy was performed at one or more segments. After a midline or paramedian durotomy, the tumor was debulked with an ultrasonic aspirator, separated from the spinal cord and nerve rootlets and released from its insertion site. If feasible, the insertion site was coagulated, corresponding to a Simpson grade II resection. After thorough irrigation, the dura was closed with watertight 6-0 monofilament running sutures. The use of additional sealants was at the discretion of the surgeon. Our surgical technique for completely ossified ventrolateral meningiomas has been described previously [32]. Spinal instrumentation was not added.

In accordance with the standard practice at our institution, all patients underwent post-operative magnetic resonance imaging (MRI) 2 months after surgery, with a clinical assessment and annual FU thereafter.

### 2.4. Intraoperative Neurophysiological Neuromonitoring

MEP and SSEP were monitored with a commercially available system (Inomed, Emmendingen, Germany). For SSEP, the median and posterior tibial nerves (PTN) were stimulated with monopolar needles on both sides, using a square wave pulse of 200 µs duration, rate of stimulation of 4.7 Hz, and intensity of 10–25 mA. Recording was done on the scalp with cork-screws located in Fz, Cz’, C3′, and C4′, derived from the international 10–20 system of electroencephalography. Different bipolar montages were used, according to the stimulated nerve and signal to noise ratio [33,34]. with 100–150 averages, sweep of 50–100 ms, low/high band pass filters of 0.5/300, respectively, and sensitivity of 0.5–5 µV, as needed. For MEP, anodal constant current stimulation was performed in the scalp with cork-screw electrodes (C1 against C2 or C3 against C4, derived from the international 10–20 system), with a train of five stimuli, interstimulus interval of 4 ms, pulse duration of 500 µs, at 0.5 Hz, with a maximum intensity of 250 mA. Recording was performed with pairs of monopolar needles separated 1–2 cm from each other over at least the abductor pollicis brevis (APB), tibialis anterioris (TA), and abductor hallucis (AH) muscles [35]. Other muscles recorded were not further analyzed in this retrospective study. The recording parameters consisted of single trials with a sweep of 100 ms, sensitivity of 300 µV/Div to 5 mV/Div, as needed, and low/high band pass filters of 0.5/2000, respectively.

### 2.5. Radiologic Assessment

Pre and post-operative MRI were retrospectively analyzed. We classified the localization of the tumor with respect to the spinal cord into three groups on axial MRI: ventrolateral, lateral, and dorsolateral. The craniocaudal, ventrodorsal, and transversal tumor size was measured at its maximum expansion in millimeters. We also determined the craniocaudal expansion by calculating the total number of affected segments defined by crossing the disk space. As an easy way of measuring the extent of spinal cord compression, we analyzed the areas of tumor and intradural space at maximum compression level on axial images and between these two a ratio (tumor-canal ratio; TCR) was calculated (Areatumor/Areaduralsac × 100). The results are shown as percentages. All measurements were performed on T2-weighted MRI (Figure 1).

### 2.6. Statistics

Statistical analysis was performed using the statistical software SPSS (IBM, version 25, Armonk, NY, USA). Descriptive data included the calculation of the mean and standard deviation (SD). Statistical significance was set at a *p*-value of <0.05.

When comparing the means of different groups (i.e., nominal versus continuous variables), the data were inspected for normality and homoscedasticity (i.e., equal variances). In the case of normality and equal variances, a one-way ANOVA was carried out. If the data were normal but heteroscedastic, Welch’s ANOVA was applied. If data were non-normal but homoscedastic, a Kruskal-Wallis test was used. If the test applied showed statistical significance, post hoc tests were carried out to determine which groups had different means: Bonferroni in the case of normal data, Games-Howell in the case of non-parametric data. For the comparison of two means, independent samples Student’s t-tests were used. In all cases the null hypothesis was that the means are the same among all the groups compared.

To determine whether two nominal or ordinal variables were independent, Fischer’s exact test was carried out if at least one cell in the cross-table contained a value lower than five, otherwise a chi-squared test was used. The null hypothesis was that the proportions of the two variables are the same. In the case of two-by-two cross-tables, the odds ratio (OR) was computed to better assess the association between the two variables. Furthermore, specificity, sensitivity, positive predictive value (PPV) and negative predictive value (NPV) were computed, whenever reasonable.

## 3. Results

### 3.1. Patient Characteristics

From January 2009 to April 2021, we operated on 86 patients with SM. Of those, 75 (87.2%) were women. The mean age was 65.7 (±14.2) years (range from 34 to 93 years). The mean BMI was 26 (±4.6) kg/m^2^. Most of the patients had an American Society of Anesthesiologists classification of II (47 of 86; 54.7%) or III (29 of 86; 33.7%). Twelve patients (14%) reported current smoking. Mean FU was 29.8 (±33.2) months. Three patients (3.5%) had neurofibromatosis type 2 (Table 1).

### 3.2. Preoperative Symptoms and Functional Impairment

In fifty-seven (66.4%) patients, the clinical examination showed a sensory deficit, whereas 47 (54.7%) patients had a motor deficit ranging from BMR 0 to 4/5 (median 4/5). Ataxic gait was present in 62 (72.1%) patients. The median preoperative mMCS and Frankel scale were II and D, respectively.

### 3.3. Radiographic Assessments

Fifteen (17.4%) tumors were localized in the cervical spine, four (4.7%) at the cervicothoracic junction, and 67 (77.9%) in the thoracic spine. No patients had a meningioma in the lumbar or sacral spine. Nineteen (22.1%) meningiomas were located ventrolaterally, 25 (29.1%) laterally, and 41 (47.7%) dorsolaterally. Most meningiomas were restricted to one segment (69/86; 80.2%), whereas 15 (17.4%) and 2 (2.3%) had an extension over two or three segments, respectively. Mean craniocaudal, ventrodorsal, and transversal tumor size was 19.6 (±9.6) mm, 11.9 (±4.3) mm and 12.4 (±3.4) mm, respectively. Mean area of the meningioma and the intradural space at maximum compression level was 121.9 (±48.8) mm^2^ and 193.9 (±59.1) mm^2^, respectively. The mean TCR was 64% (±19.8). A higher TCR was associated with the presence of motor and/or sensory deficits before surgery (*p* < 0.001) and higher preoperative mMCS (*p* < 0.001).

### 3.4. Surgical and Histopathological Data

Fifty-six (65.1%) patients had a hemilaminectomy and 29 (33.7%) a complete laminectomy. No patient received additional dorsal instrumentation of the spine. Mean estimated blood loss during surgery was 365.7 (±258.4) mL and the mean duration of surgery was 205.4 (±60.8) min. Gross total resection was achieved in 81 patients (94.2%). In most cases Simpson grade II (60/86; 69.8%) or grade III (20/86; 23.3%) resections were performed. In five patients, only subtotal resection (Simpson grade IV) was achievable.

Except for one WHO grade 2 (1.2%) tumor, all other SM (85 of 86; 98.8%) were classified as WHO grade 1. The most common subtypes were psammomatous (28/86; 32.6%), transitional (28/86; 32.6%), and meningothelial (18/86; 20.9%) meningiomas. Calcification or partial ossification was detected in 42 (48.8%) and total ossification in four (4.7%) meningiomas.

### 3.5. Post-Operative Course and Functional Outcome

Mean hospital stay was 8.7 (±3.6) days. Thirty-seven (43%) patients were discharged directly home, whereas 43 (50%) patients were discharged to rehabilitation clinics. Six (7%) patients were transferred to another regional hospital. There was no perioperative mortality in our cohort. Mean FU was 29.8 (±33.2) months.

The median pre and post-operative mMCS was II and I, respectively (*p* < 0.001) (Table 2). Fifty-seven (66.3%) patients showed an improvement of at least one grade one year after surgery (Figure 2).

Higher preoperative TCR was associated with more improvement in mMCS (*p* < 0.001) and decreases of motor and sensory deficits at FU (*p* < 0.001). Patients whose mMCS did not improve had a lower TCR (*p* < 0.001). Post-operative mMCS was not associated with TCR (*p* = 0.188). Median preoperative Frankel scale was D and improved after surgical treatment to a median score of E (*p* < 0.001).

### 3.6. Complications, Reoperations and Tumor Recurrence

Seventeen (19.8%) patients had complications associated with the surgery (Table 3). Surgical revision or reoperation during the observation period was necessary in 13 (15.1%) cases. Of those, five patients (5.8%) developed post-operative epidural hematoma, which had to be removed (one patient needed this procedure twice, leading to a permanent motor deficit); three patients (3.5%) had post-operative arachnoiditis with new symptoms requiring revision surgery, and four (4.7%) patients developed a post-operative cerebrospinal fluid leak (CSFL), which was closed in a second surgery. Two patients had a new neurologic deficit after surgery, which resolved completely during FU. One patient with a radiation induced cervical meningioma treated via hemilaminectomy developed delayed C5-palsy 2 days after surgery, which persisted over the FU period. Although spinal instrumentation was not added, there were no cases of post-operative instability or kyphosis needing reoperation.

Three (3.5%) patients developed a recurrent meningioma after a mean period of 30.7 months (range 10–42 months), which was treated surgically in one case (1.2%) and by radiotherapy in two cases (2.3%). Of those three patients, two had an initial subtotal resection according to Simpson grade IV. An incomplete tumor resection according to Simpson grade IV had a significantly higher rate of tumor recurrence (*p* = 0.014; OR 53.3; 95% CI: 3.72–764.63).

### 3.7. Intraoperative Neurophysiological Monitoring and Post-Operative Outcome

Of the 86 operations, 51 (59.3%) were performed with IOM. Of these, 13 (25.5%) had IOM warnings, corresponding to pure sensory (three cases), pure motor (six cases), and mixed sensorimotor warnings (four cases). Eight patients (15.7%) had irreversible deterioration of evoked potentials (five pure motor, three pure sensory). One of the patients with irreversible alteration of MEP had a new motor and sensory deficit immediately after surgery, which had resolved at FU. None of the patients with irreversible SSEP alteration, or with reversible MEP and/or SSEP alterations had new deficits after surgery.

### 3.8. Diagnostic Accuracy

There was no association of MEP alterations (none, reversible or irreversible) with worsening of neurological status immediately after surgery (none, any (either new motor deficit, new sensory deficit, or both)) (*p* = 0.244), worsening at FU (there was none), improvement after surgery (*p* = 0.083) and at FU (*p* = 0.188), TCR (*p* = 0.075), or location of the tumor (*p* = 0.394).

There was no association of SEP alterations with worsening of neurological status immediately after surgery (*p* = 0.200), worsening at FU (there was none), improvement after surgery (*p* = 0.607), or at FU (*p* = 0.850), or with location of the tumor (*p* = 0.777). Comparison with the TCR was not possible, due to the highly non-parametric and heteroscedastic nature of the data.

The specificity between irreversible MEP alterations and new motor deficits at FU was 0.89, and that between irreversible SEP alterations and new sensory deficits at FU was 0.93. The specificity between irreversible MEP and/or SEP alterations and any new deficits at FU was 0.84. In all three cases, the PPV was 0 and the NPV was equal to 1. The sensitivity could not be determined due to the absence of new long-term neurological deficits.

### 3.9. IOM Versus No-IOM Group

The IOM and no-IOM groups had the same mean age and the same mean TCR (Table 4). The two groups also had the same proportions of sexes, locations, levels, preoperative mMCS, and neurological deficits. Thus, the two groups were homogeneous with respect to these variables. Post-operatively, the two groups had the same proportions for Simpson grade and mMCS (Table 5).

Three patients in the no-IOM group had worsened neurological status after surgery (8.57%), two of whom had a persistent deficit at FU (5.71%). In the IOM group, one patient had a new motor deficit after surgery (1.96%), but not at FU (0%). Comparing the worsening of neurological status immediately after surgery between the two groups yielded no statistically significant difference (*p* = 0.300, OR 0.213, 95% CI 0.021–2.141). At FU, there was also no statistically significant difference in worsening of neurological status between the two groups (*p* = 0.163). In this case, the OR could not be determined because there was no worsening in neurological status in the IOM group.

### 3.10. Influence of IOM on Surgical Strategies

For three patients, an explicit statement in the surgical reports noted that IOM directly influenced the surgical strategy.

Case 1: dorsolateral T7 ossified SM. During right hemilaminectomy there was a loss of MEP and a 70% decrement in amplitude of SSEP in the lower limbs. The surgery was stopped for 12 min and blood pressure was increased. SSEP recovered up to 80% of the original amplitude, and there was a further minimal recovery of MEP in the lower limbs. The patient had an increase in motor and sensory deficit after surgery, which had resolved at FU.Case 2: lateral T10–11 SM. During laminectomy there was a loss of SSEP, which recovered after pauses of a few minutes. The patient had no motor or sensory deficits.Case 3: dorsolateral T2–3 SM. The tumor was attached to the thoracic roots possibly including Th1. Direct nerve stimulation was performed and elicited no responses. The nerve was cut to achieve complete tumor resection. The patient did not present any new post-operative deficit.

In the IOM protocols, we found 13 cases where IOM alterations were documented and reported to the surgeon. This included reversible and irreversible changes. An example case is presented below.

#### Descriptive Case

A 78-year-old female patient with an ossified SM at spinal level T3 underwent surgery in our institution (Figure 3). The TCR was 61%. Before surgery, she had sensory deficits with ataxia in the lower limbs and paraparesis (BMR 4/5; mMCS 3; Frankel scale D). Baseline neurophysiological recordings were made with median and PTN SSEP, and MEP recorded from APB, abductor digiti minimi, quadrices, TA, gastrocnemius and AH muscles on both sides (Figure 4A). During laminectomy, and persisting during dura opening, a loss of MEP occurred in the left quadriceps and in all the muscles of the right lower limb, with the exception of the AH. This was followed by a decrement of more than 50% in the amplitude of the ipsilateral PTN. Blood pressure was increased. After approximately 30 min, there was a partial recovery of the PTN (with a final decrement lower than 50% of amplitude) and all the MEP, which reappeared after higher stimulus intensity (Figure 4B,C). At FU, the patient’s neurological status had not changed from that immediately after surgery. At 11 months FU, she showed complete recovery of her preoperative sensory deficit but no change in her previous motor status.

## 4. Discussion

Our study demonstrated that surgery on SM in patients with poor neurological status before surgery led to an improvement. Microsurgical resection of SM is safe and effective with low complication and recurrence rates. IOM may be a useful adjunct, especially in challenging cases such as purely ossified or ventrolateral SM and could guide surgical strategies.

Besides maximum safely achievable tumor resection, preservation of neurological function and recovery from preoperative neurological deficits are the main goals of microsurgery [3]. Extensive SM are challenging, particularly when lesions are located ventrally to the spinal cord. Predicting post-operative neurological outcome remains difficult, and both deterioration of symptoms in up to 21% of patients and severe perioperative surgical, cardiovascular, and thromboembolic complications have been reported. Due to the natural history of SM, surgery is often performed in geriatric patients who are often less healthy and have severe comorbidities. However, the introduction of modern neuroimaging and standard microneurosurgical techniques means that these tumors can now be removed safely with low morbidity and mortality [5]. Despite variations in tumor dimensions and degree of cord compression, most patients exhibit an excellent post-operative neurological outcome, regardless of older age [12]. This is in line with our findings: nearly two-thirds of our patients showed improvements in the mMCS of at least one level and only four patients developed new neurological motor deficits after surgery (including one delayed C5 palsy), from which two recovered during the FU period.

Several factors have been associated with a poor outcome, such as psammomatous-type meningioma and WHO grade 2 or higher, invasion of arachnoid or pia mater, Simpson grade resection II and III, ventral attachment, calcifications, dural tail and T2 cord signal changes, poor preoperative functional status, and sphincter involvement. In our study, we could not detect these as risk factors for poorer outcome, although we could not check for all of them [10,12,20].

The CSFL rate after surgical resection of spinal meningiomas does not differ from that associated with other intradural pathologies [36]. In our series, the CSFL rate was 4.7%, which is in line with previous studies [37,38,39]. However, most of our resections were Simpson grade II and there was no need for an additional duraplasty, which could lead to higher rates of post-operative CSFL. Since the recurrence rates in our cohort were quite low, with only 3.5% needing reoperation or radiation therapy, the surgical technique seems justified [40]. Our findings are in line with reported recurrence rates from 1 to 8% [5,7,19,20,41,42].

Few studies have explored the relationship between tumor dimensions, cord compression, and functional outcome and/or preoperative neurological impairment, and results have been conflicting [6,10,43,44]. In our series, we found a significant correlation between preoperative spinal cord compression (measured as TCR) and functional impairment. Higher TCR was associated with a better recovery after surgery (defined as difference between pre and post-operative mMCS), which is explained by the greater preoperative impairment and the correspondingly greater scope for improvement. However, in cases of SM with incipient spinal cord compression and/or enlargement evident on radiological FU, early decompression by microsurgical resection is certainly advisable to prevent further preoperative functional impairment and facilitate gross total resection.

### 4.1. IOM

Several studies have discussed the value of IOM as a diagnostic aid during spine surgery. Applying correct alarm criteria, for example a loss in MEP, can predict transient or permanent post-operative motor deficit. In the IOM group in our series, alterations in evoked potential monitoring correctly predicted post-operative worsening in all affected patients. This is in concordance with previous reports on the use of IOM during spine surgery [45,46]. IOM has been shown to predict immediate post-operative outcome [24,25,47,48,49], whereas loss of TES MEP would indicate immediate post-operative paresis.

The imminent question is if the use of IOM could prevent intraoperative injury and improve post-operative outcome [50]. To our knowledge, there is so far no class I evidence from studies on the preventive value of IOM. The most discussed paper on this issue I probably the recent publication by Hadley et al. [51]. Following a review of the literature, the authors highlight the diagnostic value of IOM but rejecting the therapeutic value. A limitation is that Hadley et al. mixed spine surgeries (especially complex stabilization procedures), extramedullary spinal cord tumor surgeries, and intramedullary tumor surgeries. These surgeries are very different in terms of neurosurgical and neurophysiological considerations.

In our series, we were not able to demonstrate any significant association between any features of neurophysiological responses (such as MEP thresholds at baseline, presence or absence of MEP or SSEP at baseline, or MEP or SSEP changes during surgery) and preoperative patient status, tumor features, or post-operative outcome. Regarding worsening of motor status, we found that three patients in the no-IOM group and one patient in the IOM group presented motor impairment post-operatively. At long-term FU, two of the patients in the no-IOM group had not recovered, but the patient with motor impairment in the IOM group had fully recovered. Hence, even though the IOM group was larger (*n* = 51) than the no-IOM group (*n* = 35), we observed less neurological worsening in the IOM group than in the no-IOM group (directly after surgery 1.96% versus 8.57% and at FU 0% versus 5.71%). We observed a tendency towards a better motor outcome in the IOM group, but it was not statistically significant, possibly due to the small number of patients with new motor deficits. The small sample size of patients with worsening of neurological status in the IOM group (*n* = 1 immediately after surgery, *n* = 0 at FU), albeit surgically desirable, constrains the statistical analysis. Further multicenter studies with pooled data could help to overcome the limitation of low incidence of intraoperative alarm events and/or post-operative neurological worsening.

More importantly, the patients in the two groups were neither randomized nor case-matched. We retrospectively analyzed a consecutive series of patients, and the assignment to IOM was done at the discretion of the surgeon and availability of IOM. This might further limit a direct comparison between the groups.

Furthermore, we were not able to assess all reversible MEP or SSEP changes, due to documentation failure. Thus, their effect on the surgical strategy could not be definitively analyzed. However, we found three cases with detailed documentation in the operative reports and 13 cases in the IOM protocols. Thus, the change in the surgical strategy triggered by an IOM alarm might have been underestimated in our series and might have had an additional benefit for preventing neurological deficits. Furthermore, the high specificity and NPV of IOM may reassure the surgeon that it is safe to continue with the ongoing surgical maneuver without expecting neurological worsening.

### 4.2. Limitations

In addition to the considerations mentioned above, this study was subject to the following limitations: First, the Simpson grading was not consistently recorded in the surgical report, so it had to be reconstructed from the description of the surgical technique. In a few cases, Simpson grade III resections were probably Simpson grade II, but had not been adequately recorded. This might be relevant because the extent of resection often necessitates more spinal cord manipulation in certain cases, which might have a negative impact on functional outcome. Additionally, the mean FU of 30 months could have limited the information regarding tumor recurrence. Finally, the retrospective nature of the study and low numbers of new neurological deficits made the statistical analyses less informative.

## 5. Conclusions

Microsurgical resection of SM is safe and leads to neurological improvement in most patients. The rates of peri-operative and post-operative complications and tumor recurrences were low. We observed a tendency towards better neurological outcome in the IOM group, although no statistically supported preventive benefit of use of IOM in meningioma surgery was found. We recommend the use of IOM in complex cases, such as completely ossified or large ventrolateral meningioma, where some spinal cord manipulation is most likely unavoidable. Further studies with larger cohorts are required to fully investigate the preventive benefit of IOM in SM surgery.

## Figures and Tables

**Figure 1 cancers-14-03989-f001:**
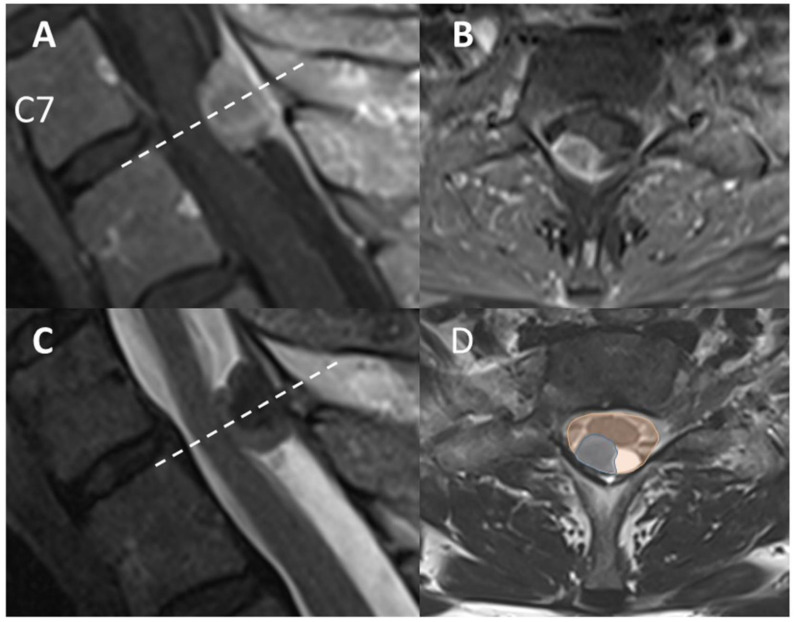
(**A**,**B**) Preoperative sagittal and axial T1-weighted contrast-enhanced MR images show an intradural extramedullary lesion at T1–2 with displacement of the spinal cord. (**C**,**D**) Preoperative sagittal and axial T2-weighted MR images. (**D**) measurement of the tumor-canal ratio (TCR), blue: cross-sectional area of tumor at level of maximum expansion, orange: spinal canal area.

**Figure 2 cancers-14-03989-f002:**
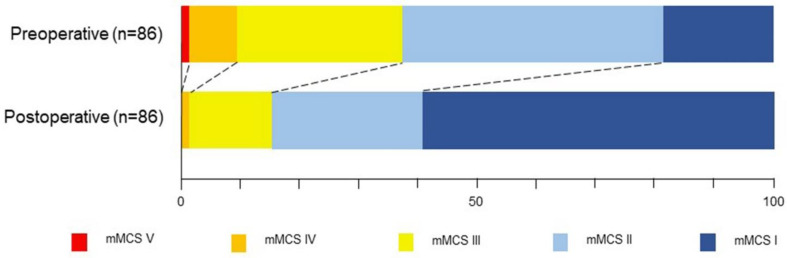
Modified McCormick Scale pre and post-operative at 1-year follow-up. Most of the patients (57/86; 66.3%) showed improvement of at least one grade. The median preoperative Modified McCormick Scale score was II whereas the median post-operative score was I (*p* < 0.001). mMCS: Modified McCormick Scale.

**Figure 3 cancers-14-03989-f003:**
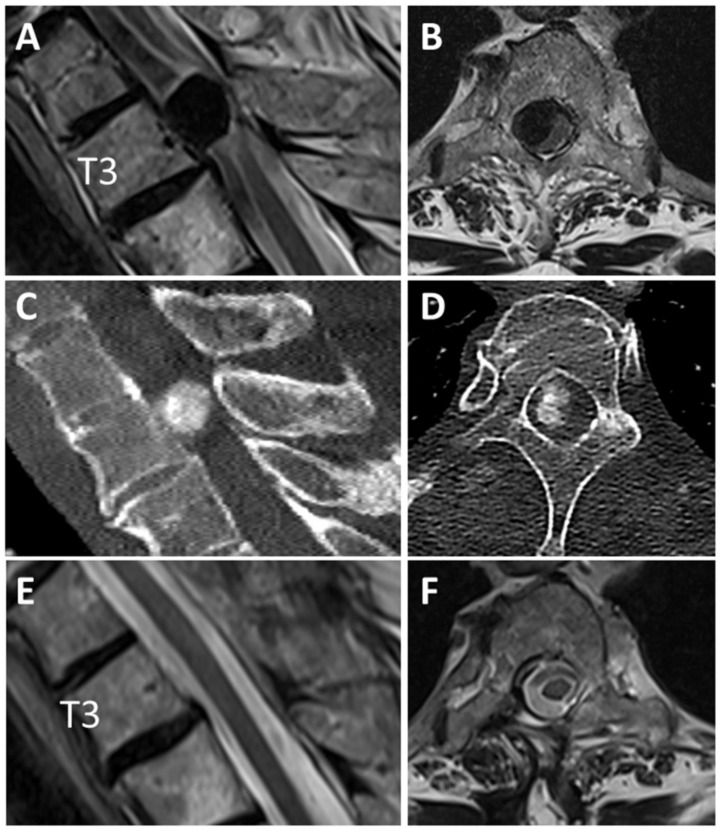
Ossified spinal meningioma at T3 of a 78-year-old woman presenting with gait disturbance and sensorimotor deficits (BMR 4/5). (**A**,**B**) Preoperative sagittal and axial T2-weighted MR images show a large intradural extramedullary lesion at T3 with displacement and compression of the spinal cord. Tumor-canal ratio (TCR) was 61%. (**C**,**D**) The CT scans revealed an ossification of the lesion. (**E**,**F**) Post-operative sagittal and axial T2-weighted MR images show complete resection with no signs of recurrent meningioma 1 year after surgery.

**Figure 4 cancers-14-03989-f004:**
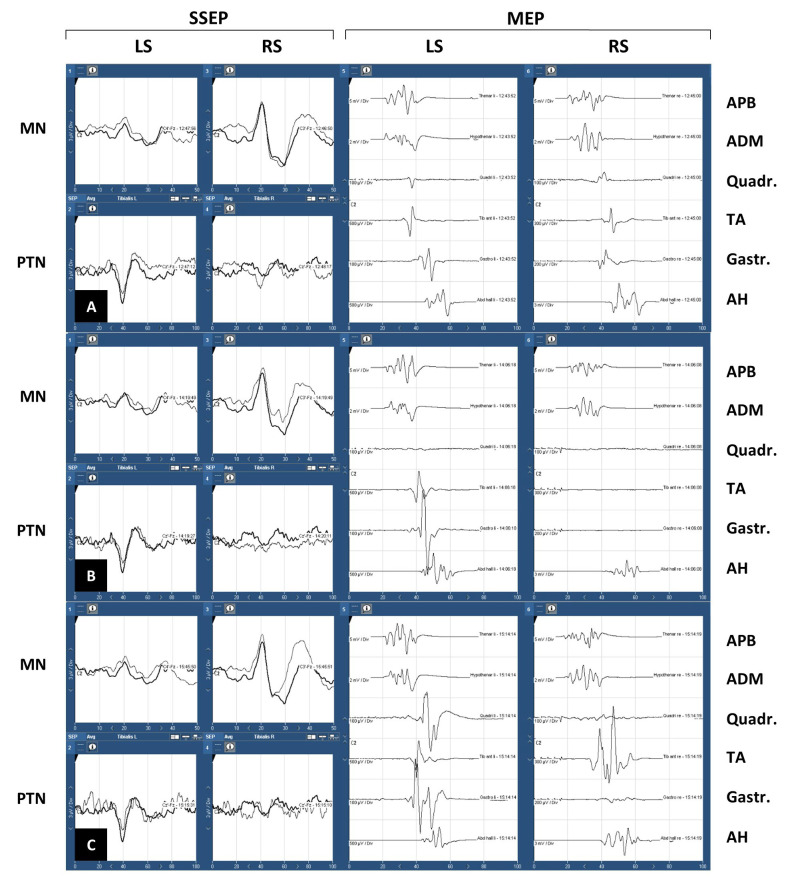
Descriptive case. (**A**) Baselines. (**B**) Loss of MEP of both Quadr and TA and Gastr in the right lower limb. Decrement of amplitude of PTN SSEP in the same limb. (**C**): Partial recovery of motor and sensory potentials. SSEP: somatosensory evoked potentials; MEP: motor evoked potentials; LS: left side; RS: right side; MN: median nerve; PTN: posterior tibial nerve; APB: abductor pollicis brevis muscle; ADM: abductor digiti minimi muscle; Quadr: quadriceps muscle; TA: tibialis anterioris muscle; Gastr: gastrocnemius muscle; AH: abductor hallucis muscle.

**Table 1 cancers-14-03989-t001:** Patient characteristics.

No. of Patients	86 (100%)
Sex	
Male	11 (12.8%)
Female	75 (87.2%)
Age, mean (±SD), years	65.7 (±14.2)
BMI, mean (±SD), kg/m^2^	26 (±4.6)
ASA	
I	8 (9.3%)
II	47 (54.7%)
III	29 (33.7%)
IV	2 (2.3%)
Current smoker	12 (14%)
Neurofibromatosis type 2	3 (3.5%)
Follow-up, mean (±SD), months	29.8 (±33.2)

ASA: American Society of Anesthesiologists classification, BMI: body mass index, SD: standard deviation.

**Table 2 cancers-14-03989-t002:** Pre and post-operative modified McCormick Scale.

Modified McCormick Scale	Preoperative	One Year Post-Operative
I	16 (18.6%)	51 (59.3%)
II	38 (44.2%)	22 (25.6%)
III	24 (27.9%)	12 (14%)
IV	7 (8.1%)	1 (1.2%)
V	1 (1.2%)	0 (0%)

**Table 3 cancers-14-03989-t003:** Complications and reoperations.

**Reoperations**	**13 (15.1%)**
Recurrent meningioma	1 (1.2%)
CSFL	4 (4.7%)
Arachnoiditis	3 (3.5%)
Post-operative epidural hematoma	5 (5.8%) *
**Other complications**	**4 (4.7%)**
New post-operative deficit directly after surgery	2 (2.3%)
delayed C5 palsy	1 (1.2%)
Pneumonia	1 (1.2%)

CSFL: cerebrospinal fluid leakage; * one patient developed two post-operative epidural hematomas and after the second, a motor deficit BMR 4/5 persisted.

**Table 4 cancers-14-03989-t004:** IOM versus no-IOM, before surgery.

	IOM	No-IOM	
Number of patients	51	35	
Age (years, mean ± SD)	66.1 ± 14	65.8 ± 13.4	*p* = 0.914
Sex			*p* = 1.000
Women	44	31	
Men	7	4	
Location			*p* = 0.147
Dorsolateral	28	13	
Lateral	15	10	
Ventrolateral	8	11	
NA	0	1 **	
Level			*p* = 0.841
Cervical	8	7	
Junction	3	1	
Thoracic	40	27	
Modified McCormick preoperative			*p* = 0.874
I	10	6	
II	22	16	
III	15	9	
IV	3	4	
V	1	0	
Tumor-canal ratio *** (TCR) (%, mean ± SD)	66.2 (±19.7) (n = 50)	61.8 (±20.5) (n = 33)	*p* = 0.328
Deficit before surgery			*p* = 0.502 *
None	10	9	
Sensory	12	8	
Motor	5	5	
Both	24	13	

* The indicated *p*-value is the result of the corresponding chi-square or Fisher’s exact test of the 2 × 2 cross-table comparing IOM (yes/no) with deficits (none/any). ** This patient did not have an MRI. *** For some patients, the TCR could not be determined due to technical limitations (e.g., bad quality MRI).

**Table 5 cancers-14-03989-t005:** IOM versus no-IOM, after surgery.

	IOM	No-IOM	
Simpson grade			*p* = 0.482
I	0	1	
II	34	26	
III	13	7	
IV	4	1	
Modified McCormick post-operative			*p* = 0.918
I	29	22	
II	13	9	
III	8	4	
IV	1	0	
V	0	0	
Deficit directly after surgery			*p* = 0.210 *
None	10	11	
Sensory	14	7	
Motor	6	4	
Both	21	13	
Deficit at follow-up			*p* = 0.232 *
None	30	25	
Sensory	8	5	
Motor	6	3	
Both	7	2	
Neurological improvement at follow-up			*p* = 0.463 *
None	23	13	
Sensory	12	7	
Motor	7	7	
Both	9	8	
Neurological worsening directly after surgery			*p* = 0.300 *
None	50	32	
Sensory	0	0	
Motor	0	1	
Both	1	2	
Neurological worsening at follow-up			*p* = 0.163 *
None	51	33	
Motor	0	2	
Sensory	0	0	
Both	0	0	

* The indicated *p*-value is the result of the corresponding chi-square or Fisher’s exact test of the 2 × 2 cross-table comparing IOM (yes/no) with the different deficits, improvement or worsening of neurological status (none/any).

## Data Availability

The datasets generated during and/or analyzed during the current study are available from the corresponding author on reasonable request.

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
