# Peer review of "Functional Outcome in Spinal Meningioma Surgery and Use of Intraoperative Neurophysiological Monitoring"

_cancers, 2022, doi:10.3390/cancers14163989_

Round 1

Reviewer 1 Report

It was my pleasure to review the manuscript about the relationship between intraoperative monitoring and neurological outcome. The par is very good written and I have only minor issue to criticize.

In the summary the paper show little novel information about the outcome after resection of spinal meningiomas. However, the authors did notice that IOM may be helpful during surgery, for example when the surgeon decides whether to resect an rootlet or to perform less radical resection.

One other critic I have, is that the authors do not focus on one primary hypothesis. Moreover, they discuss various factors that have (or do not have) influence on functional outcome. It would be better if they put the primary question IOM more in the focus.

Abstract:

calculating the mean of categorical variables such as the McCormick score it not correct in my eyes. Median would be the better choice, however, for a score between one and five, where most patients are between 1 and 3, I would abandon such calculations and leave and would only mention the absolute number and percent of patients with each score. This should be also more clear in the statistics section of “methods” and in the “results” section.

In section 3.4 (page 6) the authors mention that 23.3% of the patient had a grade III resection and defined this as gross total resection. However, most authors consider Simpson III as sub total resection. I know this a subject of definition, however, the authors should make this clear in the methods section of this manuscript.

In figure 3 the authors show the improvement of the McCormick score before and after surgery, however, in the text we notice that a minority of patient worsened after surgery. Maybe it would be possible to draw a graph showing the association between the pre and post op status in each case.

Did the patient with the c5 paly had laminectomy of hemilaminectomy? And what about the patient with the instrumentation?

For the illustrative case MR images I recommend adding the MRI images. May be instead of these in figure  2.

I the summary one more retrospective paper with a cease series on patient with spinal meningiomas. Adding a bit more information. Bigger series or registries - like those for degenerative deformities - would give us much more information and are utterly needed.

Author Response

Comment 1: Calculating the mean of categorical variables such as the McCormick score it not correct in my eyes. Median would be the better choice, however, for a score between one and five, where most patients are between 1 and 3, I would abandon such calculations and leave and would only mention the absolute number and percent of patients with each score. This should be also more clear in the statistics section of “methods” and in the “results” section.

Answer: Thank you very much for your comments and help to improve our manuscript. We also discussed this in advance. We deleted the mentioned point in the abstract and Results section. We added an according Table (Table 2) to give those numbers. (Line 29+30, 238, Table 2).

Comment 2: In section 3.4 (page 6) the authors mention that 23.3% of the patient had a grade III resection and defined this as gross total resection. However, most authors consider Simpson III as sub total resection. I know this a subject of definition, however, the authors should make this clear in the methods section of this manuscript.

Answer: We agree that the extend of resection and Simpson grading is defined differently among different authors. In our understanding, we are in line with a gross total resection defined by Simpson grade I to III. We added this definition in our methods (line 101).

Comment 3: In figure 3 the authors show the improvement of the McCormick score before and after surgery, however, in the text we notice that a minority of patient worsened after surgery. Maybe it would be possible to draw a graph showing the association between the pre and post op status in each case.

Answer: Thank you for this input. We tried to make an according figure. Since there are many patients with equal pre- and postoperative status and many patients in total, this diagram was quite confusing and we decided to keep the old figure, but added a Table (see above).

 Comment 4: Did the patient with the c5 paly had laminectomy of hemilaminectomy? And what about the patient with the instrumentation?

Answer: The patient with the c5 palsy was treated via hemilaminectomy. After carefully checking the surgical reports again, the instrumented case turned out as a mistake. So no case was instrumented in our cohort. We apologize for this mistake and changed it in our manuscript (Lines 215 and 264).

Comment 5: For the illustrative case MR images I recommend adding the MRI images. May be instead of these in figure  2.

Answer: Thank you for this comment. We deleted Figure 2 and added MRI images as Figure 3 for the descriptive case.

Reviewer 2 Report

The ms by Jesse et all elaborates on the outcome after spinal meningioma surgery with special focus on intraoperative neuromonitoring. The study is well designed and conducted, the construction of the paper clear. The subject still needs more data and the presented work gives new adds to the point. The conclusions reflect common presumptions, but the authors provide evidence for that.

My few questions/remarks would be as follows:

1. The disproportion between IOM and non-IOM group in numbers is not small. Could the authors give any background for that? Is it possible, from the side of the authors, to enlarge the non-IOM group up to ca. 50 pts?

2. I understand the role of Figure 1 (showing how to measure tumor-canal ratio), but what is the idea about Figure 2? Maybe it would be better to include pictures from "Influence of IOM on surgical strategies" or "Descriptive case"?

3. There are some punctuation, grammatical, spelling and stylistic mistakes. Needs one thorough reading. As well as editing according to journal's guidelines.

Author Response

Comment 1: The disproportion between IOM and non-IOM group in numbers is not small. Could the authors give any background for that? Is it possible, from the side of the authors, to enlarge the non-IOM group up to ca. 50 pts?

Answer: Thank you very much for your nice comment and help to improve our manuscript. In our clinical routine the standard is that spinal meningioma surgeries are performed under IOM. However, since IOM personal was not always available in clinical practice and its use rested at surgeons discretion, there are quite a few cases operated on without IOM. As the data is retrospective and all consecutive patients have been included, unfortunately we cannot enlarge the non-IOM group. Anyhow, we are planning a prospective registry.

Comment 2: I understand the role of Figure 1 (showing how to measure tumor-canal ratio), but what is the idea about Figure 2? Maybe it would be better to include pictures from "Influence of IOM on surgical strategies" or "Descriptive case"?

Answer: We deleted Figure 2 and added MRI images as Figure 3 for the descriptive case.

Comment 3: There are some punctuation, grammatical, spelling and stylistic mistakes. Needs one thorough reading. As well as editing according to journal's guidelines.

Answer: We went through the manuscript again and checked for grammatical and spelling mistakes.